# Associations Between Physical Activity, Fitness, Perceived Health, Chronic Disease and Mortality in Adult Survivors of Childhood and Young Adult Hodgkin’s Lymphoma: A Scoping Review

**DOI:** 10.3390/cancers17223625

**Published:** 2025-11-11

**Authors:** Andres Marmol-Perez, Amy M. Berkman, Kirsten K. Ness

**Affiliations:** 1Department of Physical Education and Sports, Faculty of Sport Sciences, Sport and Health University Research Institute (iMUDS), University of Granada, 18012 Granada, Spain; 2Department of Epidemiology and Cancer Control, St. Jude Children’s Research Hospital, Memphis, TN 38105, USA; amy.berkman@stjude.org

**Keywords:** exercise, childhood cancer, late effects, physical condition, strength, survivorship

## Abstract

Adult survivors of childhood and young adult Hodgkin’s lymphoma (HL) remain at a higher risk than the general population for moderate to life-threatening chronic health conditions (CHCs) including cardiac, vascular, endocrine and pulmonary impairments. Regular physical activity (PA) can improve physical fitness, reducing the risk of severity for CHCs. However, it is unclear whether adult survivors of HL experience the same benefits from PA as their peers. A scoping review was conducted to describe the associations between PA, physical fitness, perceived health, chronic disease and mortality in this population. In comparison to the general population, adult survivors of HL reported lower PA levels, had lower physical fitness and were more likely to have abnormalities in body composition and perceived health, autonomic dysfunctions and a higher risk of chronic disease and mortality than controls. An exposure to thoracic and neck radiotherapy seemed to be consistently associated with poor fitness and body composition, physical inactivity and autonomic dysfunctions.

## 1. Introduction

Advances in the diagnosis and treatment of childhood and young adult Hodgkin’s lymphoma (HL) have improved survival expectancy, with the 5-year survivorship rate exceeding 98% [1]. Unfortunately, the exposure to anticancer therapies, including radiation therapy (RT) and anthracyclines, places survivors of HL at a high risk of early onset health-related complications that increase mortality risk [2]. Survivors of HL are at a higher risk for moderate to life-threatening cardiac, vascular, endocrine and pulmonary chronic health conditions (CHCs) than their peers without a history of cancer [3].

Cardiovascular diseases are the leading non-cancer cause of death among survivors of HL [2] and are associated with exposure to thoracic and neck RT and anthracyclines [4]. After the age of 35, the incidence of coronary heart disease or congestive heart failure among adult survivors of HL treated during adolescence or early adulthood is 4- to 6-fold higher than is expected in the general population, corresponding to 857 excess cardiovascular events per 10,000 person-years [5]. The cumulative incidence of at least one grade 3–5 cardiovascular CHC by 50 years of age among survivors is 45.5% (95% Confidence Interval [CI]: 36.6–54.3), compared to 15.7% (95% CI: 7.0–24.4) in those without a history of cancer [6].

Adult survivors of HL also remain at risk of late endocrine disorders [7]. The Childhood Cancer Survivor Study (CCSS) reported that 13.3% of survivors of HL have at least one grade 3–5 endocrinopathy by the age of 35 years [7]. Those exposed to RT over 35 Gy were at the greatest risk for hyperthyroidism [8], and, when combined with chemotherapy, it increased female survivors’ risk for low ovarian reserves [9]. In addition, they often show impaired pulmonary function, including a lower forced expiratory volume, lower total lung capacity and lower carbon monoxide diffusion capacity than their peers [10]. Up to 10.1% of survivors of HL experience pulmonary dysfunction with moderate to severe symptoms by the age of 35 years [7].

Moreover, this population is also at particularly high risk for second neoplasms [11], with 9.4% of survivors of HL having at least one grade 3–5 subsequent malignant neoplasm by age 35 [7].

Adult survivors of HL often report experiencing a variety of distressing symptoms associated with treatment modalities. Among these symptoms, fatigue is particularly limiting, which includes diminished energy and mental capacity and an increased need to rest disproportionate to any recent change in physical activity (PA) [12]. This persistent fatigue, not relieved by rest, can severely impact quality of life (QoL) [13] and may impact PA level, though this relationship in young survivors of HL is complex and not fully understood [14].

Regular physical activity and the improvement of physical fitness have the capacity to reduce the risk for the early onset of CHCs in childhood cancer survivors [15]. PA is defined as any movement produced by skeletal muscles that increases energy expenditure, being not previously structured or planned [16]. Regularly meeting the minimum World Health Organization recommendations for aerobic PA, defined as ≥150 min/week of moderate activities or ≥75 min/week of vigorous activities or a combination thereof [16], is associated with lower cancer incidence [17,18,19] and recurrence [20] and a reduced mortality [17,19,20,21,22] in childhood cancer survivors. Physical fitness, which is a combination of performance in both the muscular and cardiopulmonary systems, defines one’s ability to carry out daily tasks without undue fatigue [23]. Physical fitness is considered a powerful marker of health in adulthood [24,25,26,27], and thus it is clinically relevant in follow-up care and surveillance protocols for adult survivors of HL.

Research examining the benefits of PA and physical fitness for CHCs and mortality in adult survivors of HL is needed. This scoping review aims to describe the associations between PA, physical fitness, chronic disease, mortality and perceived health (i.e., QoL and fatigue) in adult survivors of HL.

## 2. Materials and Methods

### 2.1. Search Strategy

A systematic search of published studies was carried out in MEDLINE (via PubMed), Web of Science, CINAHL and the Cochrane database of systematic reviews via OVID from inception to September 2025. The terms that we used in the search were the following: Hodgkin *, Non-hodgkin *, surviv *, body composition, lean, musc *, fat, bone, fitness, endurance, cardio *, pulmo *, strength *, “heart rate variability”, “heart rate”, “autonomic dysfunction *”, “autonomic function *”, “physical function *”, “physical capacity”, exercis *, “physical * activity”, “physical * active”, sedent *, inactiv *, fatigue, “quality of life”, quality-of-life, QOL, HRQOL, well-being and positivism (Appendix A). We also manually searched the reference lists of previous studies and retrieved potential eligible reports. The protocol was not prospectively registered in PROSPERO or any other systematic re-view registry. This review also conformed to the Preferred Reporting Items for Systematic Reviews and Meta-Analyses (PRISMA) guidelines (Appendix A).

### 2.2. Inclusion and Exclusion Criteria

Studies were eligible for inclusion in this review if they included adult survivors of HL (diagnosis at ≤39 years of age) and were cross-sectional, retrospective or prospective cohort designs. The exposure must have included either PA or fitness (i.e., muscle strength, cardiopulmonary fitness) and the outcomes included body composition, autonomic dysfunction, chronic disease/mortality, QoL or fatigue. Treatment-related risk factors for inadequate PA were also explored. We excluded studies that were not peer-reviewed, protocols and studies not written in English.

### 2.3. Screening Data Extraction

The total search retrieved 2886 records. After removing duplicates, 1567 studies remained for screening. One reviewer (AMP) initially screened the study titles and abstracts and reviewed the full texts to determine if the studies were finally eligible. Discrepancies were resolved by consensus with a second author (KKN). Finally, a narrative synthesis was conducted to summarize the included studies (Figure 1). The results from 20 primary research studies are discussed in this review.

### 2.4. Synthesis

Manuscripts were grouped by the examined outcomes including muscle strength, cardiopulmonary fitness, body composition, PA, autonomic dysfunction, associated treatment-related risk factors, associated chronic disease/mortality, QoL and fatigue. The following data were retrieved from the original reports: (1) first author; (2) country from which the data were collected; (3) year of publication; (4) sample characteristics (i.e., sex and race); (5) study design; (6) sample size; (7) age; (8) time from diagnosis/treatment completion; (9) measurements of the exposure and outcomes.

## 3. Results

### 3.1. Characteristics of the Included Studies

A total of 20 studies were included in this scoping review. Sixty percent (12/20) of the included studies focused on PA and/or fitness (i.e., muscle strength, cardiopulmonary fitness). The remaining studies reported outcomes including body composition, autonomic dysfunction, QoL and fatigue. Most studies were cross-sectional (19/20; 95%); one was longitudinal. Over half (11/20; 55%) of the studies were conducted in the United States, with three studies conducted in The Netherlands, two in Germany, two in Norway, one in Brazil and one in Turkey. A summary of the included studies is presented in Table 1.

### 3.2. Physical Activity

The evaluations of PA levels among adult survivors of HL show mixed results [28,29,30,31,39]. Wogksch et al. [39] and Williams et al. [28] observed that adult survivors of HL self-reported a similar compliance to PA guidelines as controls. In contrast, Oldervoll et al. [31] found that survivors self-reported that they were more physically active than the general population controls (48% vs. 25%). However, in this study, PA evaluation was limited to only one question. Conversely, De Lima et al. [29] showed similar PA levels reported by HL survivors vs. controls across different intensities (low: 17% [2/12] vs. 17% [6/36], medium: 25% [3/12] vs. 25% [9/36], high: 58% [7/12] vs. 58% [21/36]). Jones et al. [30] reported that the mean vigorous intensity exercise behavior was 6.1 + 6.3 MET-hours/week, with 35.9% (427/1187) of HL survivors reporting no vigorous intensity exercise behavior.

### 3.3. Fitness

Roper et al. [41] suggest that self-reported functional status improves to baseline six months after treatment completion. However, others report that muscle strength deficits are prevalent years later in adult survivors of HL [37,39,43]. Among the participants in the St. Jude Lifetime Cohort Study (SJLIFE) [39], adult female survivors of HL had lower knee extension (145.7 + 4.0 vs. 163.4 + 4.0 newton meters [Nm]/kilogram [kg]) and handgrip strength (29.8 + 0.5 vs. 32.2 + 0.5 Nm/kg) values compared to controls. In the same study, no muscle strength deficits nor muscular fatigue were found amongst survivors. However, close to 20% of male survivors had an excess impairment (compared to 6.7% expected in the general population) in muscular fatigue in comparison to age-, sex- and race-specific comparison group values (percentage of those with ≥1.5 SD). This proportion was less than 6.7% in female survivors. Van Leeuwen-Segarceanu et al. [43] reported neck muscle weakness in 31% (25/81) of adult survivors of HL. Similar findings are reported by Tacyildiz et al. [37] in CCSS, with 13% (6/46) of survivors self-reporting physical function limitations and 10% reporting difficulty in walking or extremity movements.

Deficits in cardiopulmonary fitness are reported amongst adult survivors of HL in multiple studies [33,35,36,37,39]. In a recent report by Wogksch et al. [39] both male and female adult survivors of HL enrolled in SJLIFE had a poor walking efficacy, indicated by less distance covered during the six-minute walk distance test (6MWT) (male: 604.4 + 7.9 vs. 637.7 + 7.5 m; female: 564.5 + 6.9 vs. 590.6 + 7.0 m) compared to controls. Similarly, Tacyildiz et al. [37] noted that 10% (5/50) of adult survivors of HL presented with impairments in walking, whereas no impairments were noted in a control group. Rizwan et al. [33] reported that, among adult survivors of HL evaluated for cardiopulmonary fitness via cardiopulmonary exercise testing (CPET), half (32/64) had poor cardiopulmonary fitness (relative peak VO2 peak < predicted 85%). Adam et al. [35] also noted impairments in cardiopulmonary fitness. Among adult survivors of HL who completed CPET, 19.6% (9/46) exercised for less than six minutes and 30.0% (13/43) had a decreased VO2 peak (identified as less than 20 mL/kg/m^2^). Furthermore, 39.5% (20/48) complained of shortness of breath, with 12.5% (6/48) reporting it as a significant problem.

In contrast, even in a study where 6/12 HL survivors had a reduced exercise tolerance, other measures of cardiopulmonary fitness, including maximal heart rate, respiratory rate, tidal volume, systemic arterial blood oxygen saturation, ventilatory equivalent for oxygen, cardiac output and stroke volume, were not impaired [36].

### 3.4. Body Composition

Evidence suggests that adult survivors of HL have a similar body composition, including lean and fat mass, compared to controls [39,46], yet may be at risk for a lower BMD [45,46]. For example, using Dual X-ray Absorbtiometry (DXA), Wogksch et al. [39] found similar mean body fat percentages among male (25.4 + 0.6 vs. 24.8 + 0.6%) and female (35.2 + 0.7 vs. 35.1 + 0.7%) survivors of HL and controls. However, nearly 20% of male survivors had an impaired body fat percentage in comparison to age-, sex- and race-specific comparison group values (≥1.5 SD). Similarly, in another study employing DXA among adult survivors of HL (*N* = 88), van Beek et al. did not find lean mass or BMD differences [46]. Kaste et al. [45], however, reported suboptimal BMD age- and sex-matched z-scores assessed with quantitative computed tomography in adult survivors of HL (<−1.5 Standard Deviation [SD] = 14.7%; <−2.0 SD = 7.3%). Male survivors were more likely than female survivors to have BMD z-scores < −1.5.

### 3.5. Autonomic Dysfunction

Autonomic dysregulation, identified as an abnormal resting heart rate (RHR) or delayed heart rate recovery (HRR) after an exercise bout [48], is prevalent in adult survivors of HL [32,33,39]. Wogksch et al. [39] found that male and female adult survivors of HL had a higher RHR than controls (male: 77.6 + 1.0 vs. 71.2 + 0.9 beats per minute [bpm]; female: 85.4 + 1.1 vs. 76.6 + 1.2 bpm), and Heemelaar et al. [32] also identified that 40% (30/75) of survivors had an elevated RHR (91 + 10 bpm). These data are supported in a study by Rizwan et al. [33], who showed that survivors with an abnormal (<85% predicted) VO2 peak had a higher RHR (92 vs. 82 bpm), lower peak HR (154 vs. 170 bpm) and slowed HRR when compared to those with a normal predicted VO2 peak (25% [8/32] vs. 3% [1/32]).

### 3.6. Associations of Treatment with Poor Fitness, Physical Inactivity, Body Composition and Autonomic Dysfunctions

A proportion of adult survivors of HL do not meet PA guidelines [29], do not have optimal physical fitness [33,43], do not have an ideal body composition [46] and can present with autonomic dysregulation [35,47], likely related to organ dysfunction after anticancer therapies. Van Leeuwen-Segarceanu et al. [43] showed that survivors treated with mantle field RT had a high prevalence of neck muscle weakness (83% [10/12]), abnormal neck flexor strength (67% [8/12]), abnormal neck extensor strength (33% [4/12]), abnormal shoulder abductor strength (67% [8/12]), abnormal elbow flexor strength (42% [5/12]) and abnormal antebrachial flexor strength (17% [2/12]). In addition, 100% of survivors irradiated more than 30 years ago had neck muscle weakness, whereas none had it after RT less than 10 years ago.

Rizwan et al. [33] identified a negative association between time since RT and the predicted VO2 peak, where the predicted VO2 peak was 13.2% lower for every 10 years of survival among survivors exposed to RT. Survivors with an abnormal (<85% predicted) VO2 peak were more likely to have undergone mantle and para-aortic RT (72% [23/32] vs. 41% [13/32]) and less likely to have undergone combination RT and anthracycline therapy (28% [9/32] vs. 53% [17/32]) than those with a normal predicted VO2 peak. Van Beek et al. [46] showed that female survivors of HL treated with combination chemotherapy adriamycin (doxorubicin), bleomycin, vinblastine and dacarbazine or epirubicin, bleomycin, vinblastine and dacarbazine had a higher body fat percentage (mean difference 5.4%; 95% CI: 2.12–8.65). In addition, those additionally treated with mechlorethamine, vincristine, procarbazine and prednisone had a lower total body BMD (mean difference 0.052 g/cm2; 95% CI: 0.09 to 0.02) than controls. No significant differences were found between the rest of the treatment categories and controls, neither in females nor in males.

Groarke et al. [47] reported a higher RHR (78 + 12 vs. 68 + 12 bpm), higher frequency of abnormal HR (31.9% vs. 9.3%) and lower rate–pressure product at peak exercise (25,065 + 5459 vs. 26,411 + 5794 mm of mercury [mmHg] × min) in HL survivors exposed to RT compared to controls. In the same study, a lower exercise duration was identified in those with an elevated RHR (9.2 + 2.9 min vs. 10.7 + 3.1 min) and in those with an abnormal RHR (8.3 + 3.1 min vs. 10.6 + 2.9 min). Moreover, an elevated RHR and abnormal HRR were associated with a 1.1 + 0.4 and a 1.0 + 0.4 reduction in METs achieved among survivors exposed to RT, respectively. Likewise, Adam et al. [35] observed that 57% (24/42) of survivors treated with thoracic RT had a reduced HR variability and 31% (13/42) had an increased HR (>90 bpm). Those also exposed to anthracyclines, in addition to thoracic RT, also had an increased average 24 h HR (94.8 vs. 84.7 bpm).

### 3.7. Associations of Fitness, Physical Activity and Autonomic Dysfunction with Chronic Disease and/or Mortality

Wogksch et al. [39] showed that adult survivors of HL with cardiovascular, neurologic or pulmonary chronic conditions had greater odds of having an impaired aerobic endurance compared to survivors without these chronic conditions. Rizwan et al. [33] also identified that survivors with an abnormal (<85% predicted) VO2 peak vs. normal predicted VO2 peak were more likely to have moderate to severe pulmonary disease patterns on spirometry (31% [10/32] vs. 0% [0/32]); most commonly, a pulmonary obstructive pattern was found (44% [14/32]). They were also more likely to have cardiovascular events (59% [19/32] vs. 16% [5/32]). In this study, the abnormal predicted VO2 peak was an independent predictor of future cardiovascular events (hazard ratio: 6.37; 95% CI: 2.06 to 19.80). After treadmill exercise testing, an abnormal HR, defined as a drop of 12 or 18 bpm in the first minute of active or passive recovery, was also associated with an age-adjusted increased risk for all-cause mortality, including cancer-related deaths, of 4.60 (95% CI: 1.62 to 13.02) among survivors over a median of 3 years of follow-up [47].

A report from CCSS found that the 10-year cumulative incidence of any cardiovascular event among HL survivors was 12.2% (145/1187) for those reporting 0 METs, 11.9% (141/1187) for those reporting 3–6 METs and 5.2% (62/1187) for those reporting >9 METs of PA [30]. Survivors meeting the national vigorous intensity exercise guidelines (>9 MET hours/week) had a lower incidence of any cardiovascular event, except incidences of serious arrhythmias [30]. Moreover, there was a strong inverse relationship between total vigorous intensity exercise (total MET hours/week) and the incidence of any cardiovascular event (Ptrend < 0.002) [30]. Meeting the vigorous intensity exercise guidelines was associated with an adjusted 51% decrease in the risk of any cardiovascular event in comparison with not meeting the guidelines [30]. The incidence of coronary artery disease events was also strongly inversely correlated with the total accumulation of MET hours/week (Ptrend < 0.005) [30]. Apart from heart failure events, in comparison with those reporting fewer than 9 MET hours/week, the authors found inverse relationships between meeting the national exercise guidelines (>9 MET hours/week) and the incidence of valve replacement, serious arrhythmia and cardiovascular-related mortality [30].

### 3.8. QoL and Fatigue

Survivors commonly report physical and mental factors that compromise QoL [49], which is a prognostic indicator independent of other “traditional” biomedical parameters [50]. Three reports [39,42] showed that adult survivors of HL were at risk of an impaired QoL. Specifically, using the Short-Form 36 quality-of-life instrument, physical QoL was found to be significantly lower than an adult national sample, whereas mental QoL was not different [35]. Physical QoL in those with an abnormally low VO2 peak (<20 mL/kg/m^2^) was significantly lower (47.4 vs. 54.1) than in those with a normal VO2 peak [35]. Each unit of increase in the VO2 peak was associated with an increase in the physical QoL increment of 0.60. Physical QoL was also inversely correlated with average daily HR and RHR, and an average HR greater than 90 bpm was associated with a decreased physical QoL. Williams et al. [28] described that higher PA levels were associated with a lower risk for poor general health-related QoL.

Fatigue is a distressing, persistent, subjective sense of physical, emotional and/or cognitive tiredness or exhaustion more often reported among adult survivors of HL compared to controls/siblings [29,34,35,38,40,42,43,44]. Oldervoll et al. [31] observed chronic fatigue in 30% (143/476) of adult survivors of HL, whereas Rach et al. [40] reported elevated fatigue in 17% (128/751) of survivors. Hjermstad et al. [44] observed in survivors vs. controls higher mean scores of self-reported physical (9.6 + 3.9 vs. 7.9 + 3.1), mental (5.0 + 1.8 vs. 4.3 + 1.4), total (14.6 + 5.1 vs. 12.2 + 3.9) and chronic fatigue (30% vs. 11%), using the Short-Form 36 quality-of-life instrument. Adams et al. [35] indicated that 67% (32/48) of adult survivors of HL reported feeling fatigued (35% [17/48] moderate to severe) and that lower VO2 peak values correlated with greater fatigue. Ng et al. [34] showed that 37% (185/506) of survivors had fatigue scores lower than members of the general population. In this study, a lower exercise frequency was associated with higher fatigue. Van Leeuwen-Segarceanu et al. [43] showed that 83.3% (10/12) of survivors treated with mantle field RT presented with fatigue in the neck muscles.

## 4. Discussion

The five-year survival following a diagnosis of HL is excellent. However, survivors remain at a high risk for chronic disease after treatment [2]. This review shows that adult survivors of HL have an increased risk for lower PA, poor physical fitness, low bone mineral density, autonomic dysregulation, fatigue and poor QoL compared to controls. These impairments are associated with treatment exposures and are related to chronic disease and mortality. Intervention studies are needed to determine if improving these impairments can decrease the premature risk of chronic health conditions in adult survivors of HL.

### 4.1. Possible Mechanisms for Impairments and Chronic Health Conditions in Adult Survivors of Childhood and Young Adult Hodgkin’s Lymphoma

The treatment for HL can damage the organ systems needed to support regular PA. An early exposure to RT likely stimulates reactive oxygen species and may result in an inflammatory response that alters pulmonary, cardiac, peripheral vascular and muscle structure and function [51]. Mediastinal RT is associated with pulmonary fibrosis, which limits lung expansion and gas exchange [52,53]; direct cardiotoxicity, which can reduce stroke volume and cardiac output, limiting oxygen delivery [54,55]; and vascular injury through coronary artery disease or vascular stiffening affecting blood flow [56]. Childhood cancer survivors treated with >30 Gy of chest RT perform poorly on cardiopulmonary exercise testing [48,57]. RT directly damages muscles and stimulates an excessive differentiation of fibroblasts into myofibroblasts. RT-exposed muscle is characterized by fibrosis as a result of increased collagen deposition, poor vascularity and scarring [58]. Mantle field RT can cause local muscle weakness [43] and impacts performance in cardiopulmonary exercise testing [33]. A suboptimal body composition may also influence PA participation [46]. During therapy, corticosteroid exposures stimulate appetite and can promote weight gain [59], while therapy-related fatigue promotes inactivity and muscle loss [60]. Post therapy, these habits are compounded by general population risk factors (i.e., unhealthy diet, lack of exercise, smoking) and genetic susceptibility [61].

For some HL survivors, PA may also be uncomfortable because of autonomic dysfunction [35,47]. Among adult survivors of childhood cancer, cardiac autonomic dysfunction is associated with exercise intolerance, suggesting that survivors have both a blunted autonomic response to and difficulty restoring parasympathetic tone after exercise [48,57]. Attenuated blood pressure, blunted heart rate responses and a delayed heart rate recovery in response to exercise testing are common in survivors treated with chest or neck RT [62], where sensitive structures including the vagus nerve and the carotid sinus may be exposed [47]. Reports of autonomic dysfunction among survivors exposed to anthracyclines are not as consistent. One small study from Adams et al. reported a higher average 24 h RHR in adult survivors of HL [35], while Groarke et al. found no significant associations between anthracycline exposure (by dose) and an elevated RHR or abnormal HRR after exercise [47]. Additional research is needed to understand the mechanisms of autonomic injury so that interventions can be developed to manage this impairment. With the introduction of emerging treatments such as checkpoint inhibitors, it will also be essential to carefully track how these therapies influence late effects. Given that many of the late effects have been linked to radiation, fewer side effects could be observed after a reduced exposure to radiation. Nevertheless, it remains uncertain what new late effects might appear as a result of these new therapies. This highlights the need to closely monitor patients receiving these emerging treatments.

### 4.2. Possible Mechanisms for PA, Fitness, Perceived Health, Chronic Disease Associated with Mortality in Adult Survivors of Childhood and Young Adult Hodgkin’s Lymphoma

Fitness impairments and inactivity can place adult survivors of HL at an increased risk for the early onset of chronic diseases [33,39], which altogether can ultimately lead to premature mortality [30,47] driven by multiple interconnected pathophysiological processes. Insufficient PA and reduced fitness levels reduce insulin sensitivity, foster chronic low-grade inflammation and impair endothelial and arterial wall function [63]. This can contribute to obesity and dyslipidemia—factors that collectively heighten cardiovascular disease risk and mortality [63]. Evidence from the CCSS shows that the cumulative all-cause mortality risk is significantly higher among those accumulating lower PA levels (11.7% among survivors engaging in 0 MET-h/week of exercise, compared with 8.6% for 3–6 MET-h/week, 7.4% for 9–12 MET-h/week and 8.0% for 15–21 MET-h/week) [30]. In contrast, PA can suppress systemic chronic proinflammatory/reactive oxygen species (ROS) through an increased expression of endogenous antioxidant enzyme machinery, in addition to improving traditional cardiovascular risk factors (i.e., abnormal lipids, hypertension and metabolic dysregulation) [64,65]. However, 50–80% of adolescent and young adult survivors of childhood cancer fail to meet current PA recommendations [66], which can predispose them to an increased risk of mortality. Consequently, future studies should focus on developing and sustaining effective PA interventions tailored to adult HL survivors, considering their unique treatment histories and potential physical limitations [67].

### 4.3. Interventions to Decrease the Risk of Premature Chronic Health Conditions and/or Mortality

Given that survivors of childhood and adolescent HL are at a high risk of cardiovascular disease and mortality [2,3], and that PA appears to influence this risk, interventions are needed to promote an active lifestyle. Multidisciplinary approaches are likely most effective [67], combining interventions such as aerobic training, resistance training, cognitive behavioral therapy, mind–body therapy and dietary plans. Pharmacological strategies to address underlying impairments could be synergistic.

Aerobic training is considered as any activity in which the body’s large muscles move in a rhythmic manner for a sustained period of time (i.e., walking, running, swimming, bicycling) [16]. Exercise interventions in adults without a cancer history are associated with a reduced risk of coronary heart disease, stroke, hypertension, diabetes, metabolic syndrome and mortality [68]. Aerobic training improves myocardial damage from ischemia/reperfusion injury [69], promotes left ventricular remodeling and imparts modest improvements in the ejection fraction in patients with congestive heart failure [70,71]. Aerobic training also improves diastolic function [72], enhances endothelium-dependent vasodilation [73,74], increases production of nitric oxide [75] and improves baroreflex function [76]. In adult survivors of lymphoma, Courneya et al. [77] randomized 122 participants (22 with HL) in a 12-week supervised cycling program, reporting significant gains in physical functioning, VO2 peak (+5.2 mL/kg/min, *p* < 0.001) and lean body mass. A secondary analysis showed that survivors who received aerobic training had a higher progression-free survival at 61 months (68.5%) compared to controls (59%) [78]. Aerobic training appears to be safe and may improve cardiorespiratory fitness while reducing cardiovascular disease risk in HL survivors.

Resistance training improves neuromuscular strength, power and muscle mass [79]. Muscle impairment predisposes HL survivors to the premature development of traditional cardiovascular risk factors such as hyperlipidemia, hypertension and type II diabetes [51]. Skeletal muscle is a major site for insulin-stimulated glucose uptake, and reduced muscle function and/or mass can lead to a decreased glucose utilization and increased insulin resistance, promoting type II diabetes [80]. Muscle weakness also correlates with impaired vascular function and increased arterial stiffness, which ultimately contribute to the pathogenesis of hypertension [80]. Muscle activity enhances the expression of enzymes such as lipoprotein lipase, which facilitates the breakdown and clearance of triglycerides [81]. There is limited data describing resistance training interventions in survivors of HL. However, evidence suggests that structured, progressive and supervised resistance training, preferably at a moderate to high intensity [30], can help to remediate physical function impairments, prevent chronic disease and ultimately reduce the risk for all-cause mortality in other cancer survivor populations [82].

Cognitive behavioral therapy is a patient-centered form of psychotherapy that focuses on the interconnection between thoughts, feelings and behaviors, and may help survivors identify and challenge distorted cognitions and develop healthier behavioral responses [83]. Mind–body interventions are practices that aim to improve physical health by harnessing the relationship between the mind and body [84]. These interventions may include techniques such as meditation, yoga and relaxation therapy, which are designed to reduce stress and enhance well-being [84]. While specific evidence supporting these interventions in HL survivors is lacking, cognitive behavioral therapy improves sleep hygiene [85], reduces proinflammatory cytokines (i.e., IL-6, TNF-α) and improves immune function in other cancer survivor populations. Mind–body interventions also reduce systemic inflammation [86].

Although data specific to HL survivors are lacking, an unhealthy diet is an established modifiable risk factor for obesity, cardiovascular disease burden and mortality risk [87], making dietary strategies a key component of multidisciplinary lifestyle interventions. Dietary plans including plant-based foods, healthy fatty acids and a moderate consumption of animal-based foods, sugar and salt have been linked with lower risks of diet-related cardiovascular disease in people with obesity, diabetes, hyperlipidemia and hypertension [88,89,90,91]. For example, in postmenopausal women with breast cancer receiving letrozole, a telephone-based weight loss intervention reduced body weight (−4.3 vs. −0.6 kg or −5.3% vs. −0.7% at 6 months; −3.1 vs. −0.3 kg or −3.6% vs. −0.4% at 24 months) and occurred consistently across strata (BMI from 24 to <30 vs. ≥30 kg/m^2^; prior vs. no prior adjuvant chemotherapy) [92]. Pharmaceutical strategies are available to support dietary interventions and promote weight loss, including glucagon-like peptide-1 (GLP-1) receptor agonists and glucose-dependent insulinotropic peptide and GLP-1 receptor co-agonists [93,94,95,96].

Given that both healthy and unhealthy behaviors tend to occur together rather than independently [97,98], strategies that target multiple aspects of lifestyle simultaneously may be the most effective approach among adult survivors of HL. This population presents unique deficits and limitations, and hence further research is necessary to determine the most effective strategies for managing these challenges.

## 5. Conclusions

Adult survivors of childhood and young adult HL who meet PA guidelines and/or are physically fit have improved markers of cardiovascular risk factors, mortality, fatigue and QoL compared to less active/fit survivors. Anticancer treatments are known to impact organ function, reducing physiologic reserves and associated exercise responses. Research is needed to determine if and which specific exercise interventions (assessing the most appropriate frequency, intensity, time, type, volume and progression of the exercise prescription) best contribute to cardiovascular adaptations and maximize health benefits in adult survivors of HL. However, this population should be encouraged to be as active as possible through adapted programs that address the particular physiological features of this population.

## Figures and Tables

**Figure 1 cancers-17-03625-f001:**
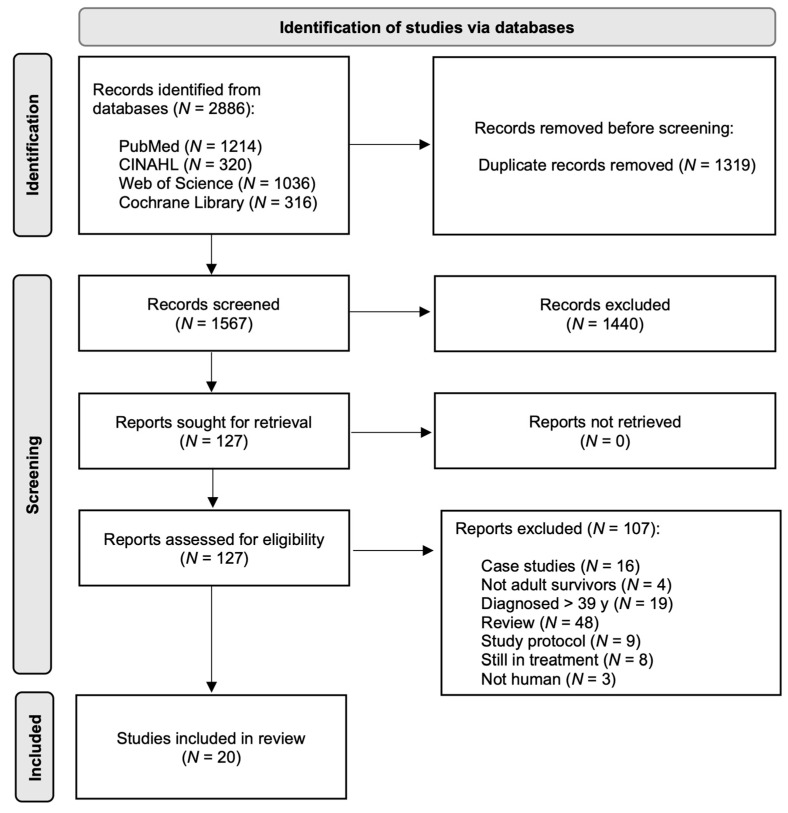
Manuscript selection flowchart.

**Table 1 cancers-17-03625-t001:** Study characteristics.

Author Country	Population	Age at Evaluation Time from Diagnosis/Treatment	Treatment Exposure	Outcomes of Interest
Physical activity				
Williams et al., 2022 [28] United States	*N* = 1760 47.9% male 91.8% white	37 (21–54) 23.15 (14.7, 34.3) median years from diagnosis	RT: 79.1% Anthracycline: 43.4% Cyclophosphamide: 59% Methotrexate: 6.2% Bleomycin: 35.8% Corticosteroids: 58.1%	=PA guidelines compliance PA was associated with lower risks in general QoL domains (RR: 0.64 [95% CI: 0.55–0.75]) ↑ RR of impairment in all QoL domains, except for pain
De Lima et al., 2018 [29] Brazil	*N* = 512 41.7% male 25.0% white	32 ± 8 4.8 ± 3.54 mean years from diagnosis	NR	↑ Fatigue among survivors than controls (14.0 ± 3.9 vs. 10.8 ± 3.4, *p* = 0.009) =PA levels among survivors and controls
Jones et al., 2014 [30] United States	*N* = 1187 50.5% male 85.3% white	42 (22–58) 16.7 (8.2–28.7) mean years from diagnosis	Any RT: 91.4% Chest RT: 85.7% Any chemo: 63% Alkylating agent: 59.8% Anthracycline: 20%	Mean vigorous intensity exercise behavior was 6.1 ± 6.3 METs hours/week^−1^ 36% of survivors reported no vigorous intensity exercise behavior The cumulative incidence of any cardiovascular event (*p* < 0.001) was as follows: ○ 12.2% at 10 years for survivors reporting 0 METs ○ 11.9% for those reporting 3–6 METs ○ 5.2% for those reporting >9 METs Survivors meeting national vigorous intensity exercise guidelines (≥9 MET hours/week^−1^) had lower incidence of any cardiovascular event, except incidence of serious arrhythmias There was a strong, graded inverse relationship between total vigorous intensity exercise (total MET hours/week^−1^) and incidence of any cardiovascular event (Ptrend = 0.002). Compared to survivors reporting 0 MET hours/week: ○ Survivors with 3 to 6 MET hours/week showed adjusted rate ratio of 0.87 (95% CI, 0.56 to 1.34) ○ Survivors with 9 to 12 MET hours/week showed adjusted rate ratio of 0.45 (95% CI, 0.26 to 0.80) ○ Survivors with 15 to 21 MET hours/week showed adjusted rate ratio of 0.47 (95% CI, 0.23 to 0.95) Adherence to guidelines for vigorous intensity exercise (i.e., >9 MET hours/week) was associated with an adjusted 51% decrease in the risk of any cardiovascular event in comparison with not meeting the guidelines (<9 MET hours/week) (*p* = 0.002) There was a strong graded inverse relationship between total MET hours/week and incidence of coronary artery disease events (Ptrend = 0.005) With the exception of heart failure events and compared with those reporting fewer than 9 MET hours/week^−1^, there were inverse relationships between meeting the national exercise guidelines (≥9 MET hours/week^−1^) and incidence valve replacement, serious arrhythmia and cardiovascular-related mortality
Oldervoll et al., 2007 [31] Norway	*N* = 476 56.0% male	46 (21–73) 17 ± 7.5 median years follow-up	RT: 27% Chemo: 18% Combined: 53%	↑ Physically active than general population (48% vs. 25%, *p* < 0.0001) 30% reported chronic fatigue
Fitness: cardiorespiratory				
Heemelaar et al., 2021 [32] The Netherlands	*N* = 75 41.3% male	46 ± 12 17.6 (12.4–25.8) mean years from treatment	RT: 100% Chemo: 80%	40% had elevated RHR (91 ± 10) defined as >80 bpm =Baseline characteristics between patients with and without elevated RHR
Rizwan et al., 2021 [33] United States	*N* = 64 33.0% male 100.0% white	51 (26–71) 23 (11–51) median years from treatment	Mantle RT: 25% Mantle and para-aortic RT: 55% Mini-mantle RT: 1.6% Mantle and cardiac RT: 6.3% Anthracyclines: 41% Surgery: 64%	50% of survivors had abnormal predicted VO_2_ peak of <85% Survivors exercised for a median of 7.6 METS, with a predicted VO_2_ peak of 26.4 cc/min/m^2^ ↓ Predicted VO_2_ peak by 13.2% every 10 years for survivors treated with RT (95% CI: −20.1% to −6.3%, *p* < 0.001) Significantly negative association between predicted VO_2_ peak and time since RT (least squares mean = −7.0%, 95% CI: −11.7% to −2.4%, *p* = 0.003) Survivors with abnormal (≤85%) predicted VO_2_ peak vs. normal predicted VO_2_ peak were as follows: ○ ↑ Likely to have undergone mantle and para-aortic RT (73% vs. 41%; *p* = 0.009) ○ ↓ Likely to have undergone combination RT and anthracycline therapy (28% vs. 53%; *p* = 0.042) ○ ↑ RHR (92 vs. 82 bpm; *p* = 0.001) ○ ↓ Peak HR (154 vs. 170 bpm; *p* < 0.001) ○ ↑ Abnormal HRR (25% vs. 3%; *p* = 0.026) ○ ↓ Lower FEV1 (median 2.16 L vs. 3.14 L; *p* < 0.001) and predicted FEV1 (median 82% vs. 95%; *p* = 0.001) ○ ↑ Likely to have moderate to severe pulmonary disease patterns on spirometry (31% vs. 0%; *p* = 0.001), most commonly obstructive findings (14 of 32; 44%) ○ ↑ Cardiovascular events (59% vs. 16%; *p* < 0.001) ○ After adjusting for hypertension and BMI, abnormal predicted VO_2_ peak was an independent predictor of future cardiovascular events (HR: 6.37; 95% CI: 2.06 to 19.80; *p* = 0.001)
Ng et al., 2005 [34] United States	*N* = 503 49.0% male	44 (16–82) 15 median years follow-up	RT: 61% Chemo: 4% Combined: 35%	37% had fatigue scores worse than US general population (*p* = 0.01) ↓ Exercise frequency was significantly associated with higher fatigue (*p* = 0.03)
Adams et al., 2004 [35] United States	*N* = 48 48.0% male 94.7% white	31.9 (18.7–49.5) 14.3 (5.9–27.5) median years from diagnosis	RT: 100% Chemo: 43.75% Anthracycline: 8.33%	19.6% exercised for less than 6 min in exercise stress test 53.0% had an abnormality on their exercise stress test 30.0% had a VO_2_ peak during exercise less than 20 mL/kg/m^2^ 39.5% complained of shortness of breath, but only 12.5% reported it as a significant problem 27.0% had blunted blood pressure/HR response to exercise stress test Anthracycline exposure had a higher average 24 h HR (94.8 vs. 84.7 bpm, *p* = 0.0079) Physical QoL in those with abnormally low VO_2_ peak ≤20 mL/kg/m^2^) was 47.4, compared with 54.1 in those with a normal value (*p* = 0.011) Each unit increase in VO_2_ peak was associated with physical QoL increment by 0.60 (*p* = 0.0002) The physical QoL was inversely correlated with average daily HR and RHR (*p* = 0.002) Average HR greater than 90 bpm was associated with a decreased physical QoL (*p* = 0.0012) ↓ Physical QoL by three points compared to an adult national sample (*p* = 0.0085) =Mental QoL compared to an adult national sample VO_2_ peak was correlated with greater shortness of breath problems and fatigue (r = −0.353, *p* = 0.02) 67% reported feeling fatigued (35% stating moderate to severe)
Kadota et al., 1988 [36] United States	*N* = 12 58.3% male	NR 9.8 mean years from treatment	RT: 100% Chemo: 33.3%	50% of survivors had reduced exercise tolerance, which was manifested by subnormal values for duration of exercise or maximal oxygen uptake Maximal HR, systemic arterial blood pressure, respiratory rate, tidal volume, systemic arterial blood oxygen saturation, ventilatory equivalent for oxygen, cardiac output and stroke volume were normal in all survivors
Fitness: muscle strength				
Tacyildiz et al., 2024 [37] Turkey	*N* = 50 66.0% male	21 ± 6 0–5 years from diagnosis: 25% 5–10 years from diagnosis: 26% >10 years from diagnosis: 49%	RT: 100% Chemo: 100%	↓ Physical functioning in 14% of survivors 11% of survivors recorded impairments in pulmonary function test, whereas no participants in the control group had such impairments ↓ Difficulty in walking or extremity movements in 10% of survivors
Stadtbaeumer et al., 2020 [38] Germany	*N* = 3596 45.9% male	35 ± 11 NR	NR	↑ Overall level of cancer-related fatigue is accompanied by ↓ overall level in each functional health domains Earlier within-person deviations in cancer-related fatigue negatively predicted subsequent within-person deviations in all five functional health domains
Wogksch et al., 2019 [39] United States	*N* = 336 53.6% male 85.1% white	NR 28.1 ± 9.2 mean years from treatment	Any RT: 97.6% Lung RT (>25 Gy): 64% Chest RT (>25 Gy): 86% Bleomycin: 25% Alkylating agent: 68.5% Doxorubicin: 64.3% Dexamethasone: 2.7% Prednisone: 55.7% Vinblastine: 67.3% Vincristine: 66.7%	↓ Muscular strength only in females vs. controls (knee extension, 145.7 ± 4.0 vs. 163.4 ± 4.0 Nm/kg and handgrip strength, 29.8 ± 0.5 vs. 32.2 + 0.5 Nm/kg) Male survivors had excess impairment (higher than 6.7% in the general population) in muscular fatigue (~20%) ↓ 6MWT in males and females vs. controls (males: 604.4 ± 7.9 vs. 637.7 ± 7.5 m; female: 564.5 ± 6.9 vs. 590.6 ± 7.0 m) ↑ RHR in males and females vs. controls (male: 77.6 ± 1.0 vs. 71.2 ± 0.9 bpm; female: 85.4 ± 1.1 vs. 76.6 ± 1.2 bpm) ↑ Resting peripheral nervous system integrity scores in males and females vs. controls (adjusted mean difference in males: 1.4 [99% CI: 0.9, 2.1] and females: 1.7 [99% CI: 1.1, 2.4]) =PA guidelines compliance (odds ratio: 0.78, 99% CI: 0.56, 1.09) 20% of male survivors had excess impaired body fat percentage (≥1.5 SD from the control mean) Survivors with cardiovascular chronic conditions had greater odds of having impaired aerobic endurance (odds ratio: 2.36, 99% CI: 1.00, 5.61) Survivors with a neurologic chronic condition had greater odds of having impaired aerobic endurance (odds ratio: 2.96, 99% CI: 1.28, 6.69), muscular strength (odds ratio: 2.94, 99% CI: 1.24, 6.96), balance (odds ratio: 2.56, 99% CI: 1.13, 5.79) and peripheral nervous system integrity (odds ratio: 2.23, 99% CI: 1.06, 4.69) compared to survivors without a neurologic chronic condition Survivors with pulmonary chronic conditions had greater odds of impaired aerobic endurance (odds ratio: 2.78, 99% CI: 1.30, 5.94), compared to survivors without pulmonary chronic conditions Survivors had a mean (SD) Physical QoL of 48.1 (11.2) and a mean Mental QoL of 46.8 (11.8)
Rach et al., 2016 [40] United States	*N* = 751 49.5% male	NR NR	Chest RT (≥30 Gy): 59.3% Anthracycline: 21% Alkylating agents: 55.8% Bleomycin: 19.6% Vinca alkaloids and heavy metals: 55.7%	17% of survivors reported elevated fatigue
Roper et al., 2013 [41] United States	*N* = 40 40.0% male 90.0% white	31 ± 6 NR	RT: 5% Chemo: 59% Combined: 36%	↑ Functional status (SF-12) from end of treatment to 6 months (*p* < 0.001) Physical distress (measured via symptom distress scale [SDS]) 13 item scale → includes pain, fatigue, outlook, breathing: lower mean scores at all timepoints vs. baseline (*p* ≤ 0.0001)
Calaminus et al., 2014 [42] Germany	*N* = 725 45.9% male	NR 15.26 ± 5.89 mean years from diagnosis	RT (>30 Gy): 31.6% Chemo (>6 cycles): 28.7%	↓ QoL in all domains (physical, emotional, social, role and cognitive functioning and global) ↑ Fatigue among survivors
Van Leeuwen-Segarceanu et al., 2012 [43] The Netherlands	*N* = 81 50.0% male	53 ± 10 NR	RT: 100% Chemo: 50%	31% had neck muscle weakness Survivors were particularly disabled in the following activities: ○ Walking for more than 30 min ○ Swimming ○ Lifting the head while in a supine position, ○ Driving a car ○ Working on the computer ○ Reading ○ Lifting heavy bags 100% of survivors irradiated more than 30 years ago had neck muscle weakness (4/4), whereas none (0/13) had it after RT less than 10 years ago Among survivors treated with mantle field RT: ○ 85.0% had neck muscle weakness ○ 67.0% had abnormal neck flexors strength ○ 33.0% had abnormal neck extensors strength ○ 67.0% had abnormal shoulder abductor strength ○ 42.0% had abnormal elbow flexors strength ○ 17.0% had abnormal antebrachial flexors strength ○ 83.8% presented fatigue in the neck muscles
Hjermstad et al., 2006 [44] Norway	*N* = 475 56.0% male 100.0% white	46 ± 12 16.3 (4.4–36) median years median follow-up	RT: 31% Chemo: 14% Combined: 55%	↑ Physical, mental, total and chronic fatigue among survivors (all *p* < 0.001) ↑ Chronic fatigue three times higher than controls (*p* < 0.001)
Body composition				
Kaste et al., 2009 [45] United States	*N* = 109 50.5% male 85.3% white	NR 9.4 (5.1–13.0) median years from treatment	Lumbar spine RT: 29.3% Procarbazine: 60.2% Cyclophosphamide: 63.9% Methotrexate (>150 mg/m^2^): 30.8% Prednisone (>2000 mg/m^2^): 34.6%	↓ BMD (<−1.5 SD = 14.7%; <−2.0 SD = 7.3%) compared to matched population for age and sex All survivors with BMD Z-scores below −1.5 were white Male survivors were more likely than female survivors to have BMD Z-scores <−1.5 (*p* < 0.04)
Van Beek et al., 2009 [46] The Netherlands	*N* = 88 63.6% male	27 (18–43) 15.5 (5.6–30.2) median years from treatment	RT: 19.8% Chemo: 100%	=Total body lean mass (*p* > 0.05) =BMD of the total body and BMD and BMAD of the lumbar spine were comparable to controls in all male survivors treated with chemotherapy only ↓ Total body BMD of female survivors treated with combination chemotherapy adriamycin (doxorubicin), bleomycin, vinblastine and dacarbazine or epirubicin, bleomycin, vinblastine, dacarbazine with mechlorethamine, vincristine, procarbazine and prednisone (mean difference 0.052 g/cm^2^; 95% CI: 0.09 to 0.02, *p* = 0.006) ↑ Total body fat (%) of female survivors treated with combination chemotherapy adriamycin (doxorubicin), bleomycin, vinblastine and dacarbazine or epirubicin, bleomycin, vinblastine, dacarbazine (mean difference 5.4% [95% CI: 2.12–8.65], *p* = 0.001)
Autonomic dysfunction				
Groarke et al., 2015 [47] United States	*N* = 263 46.0% male	50 ± 11 19 (12–26) mean years from treatment	RT: 100% Adjuvant anthracycline: 46%	↑ RHR in survivors exposed to RT compared to matched controls (78 ± 12 vs. 68 ± 12 bpm) (*p* < 0.0001) ↑ Frequency of abnormal HR in survivors exposed to RT compared to matched controls (31.9% vs. 9.3%) (*p* < 0.0001) ↓ Rate-pressure product at peak exercise among survivors exposed to RT compared to matched controls (25,065 ± 5459 vs. 26,411 ± 5794) (*p* = 0.002) ↓ Exercise duration in survivors exposed to RT with elevated RHR (9.2 ± 2.9 min vs. 10.7 ± 3.1 min) and those with abnormal HRR (8.3 ± 3.1 min vs. 10.6 ± 2.9 min) (*p* < 0.0001) Elevated RHR and abnormal HR recovery were associated with a 1.1 ± 0.3 min (*p* = 0.001) and a 1.0 ± 0.4 min (*p* = 0.006) reduction in exercise duration ↑ RHR and abnormal HR recovery were associated with a 1.1 ± 0.4 (*p* = 0.002) and a 1.0 ± 0.4 (*p* = 0.007) reduction in METs achieved among survivors exposed to RT Abnormal HRR was associated with increased all-cause mortality (age-adjusted hazard ratio: 4.60 [95% CI: 1.62 to 13.02])

All studies were cross-sectional, except for Roper et al. Abbreviations: 6MWT: six-minute walking test, BMAD: bone mineral apparent density, BMD: bone mineral density, BMI: body mass index, bpm: beats per minute, CI: Confidence Interval, Chemo: chemotherapy, FEV: forced expiratory volume, HL: Hodgkin’s lymphoma, HRR: heart rate recovery, MET: metabolic equivalent of task, Nm: newton meters, NR: not reported, PA: physical activity, QoL: quality of life, RH: heart rate, RHR: resting heart rate, RR: relative risk, RT: radiation, US: United States, VO_2_: volume of oxygen consumption.

## Data Availability

The data supporting the conclusions of this review can be made available by the authors.

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
