# Peer review of "Associations Between Physical Activity, Fitness, Perceived Health, Chronic Disease and Mortality in Adult Survivors of Childhood and Young Adult Hodgkin’s Lymphoma: A Scoping Review"

_cancers, 2025, doi:10.3390/cancers17223625_

Round 1

Reviewer 1 Report

Comments and Suggestions for Authors

This is an excellent, very thorough review of physical activity and associated fitness and chronic health conditions in survivors of childhood/adolescents with Hodgkins lymphoma.  The authors have done an extensive survey of the literature in this area and summarized it in a meaningful manner for readers particularly highlighting the main points.

My only comment for the authors is to consider  incorporation in the discussion about the future with the new agents such as checkpoint inhibitors and that it will be important to monitor how this affects late effects particularly as related to physical activity and related chronic health conditions.  Looking towards the future, since many of the late effects in this category of physical activity seemed to be related to radiation, the hope will be that with decreased use of radiation this will result in a decrease in some of the physical activity late effects noted in this review.  However it is an unknown what other late effects may emerge.  This underscores the importance for the future to monitor these patients receiving these newer therapies.

Author Response

Response: Thank you. We agree with the reviewer and hence, we have included this important topic in the discussion (pages 13-14).

Reviewer 2 Report

Comments and Suggestions for Authors

This is an important paper addressing significant late complications resulting from radiation therapy and chemotherapy administered for Hodgkin lymphoma.

I will only point out the following minor issues:

  1. The 13th line of the Abstract and the first line of Section 2.3. Screening data extraction on page 3 state that 2,527 papers were included, but Figure 1 shows 2,886 papers were extracted at first, creating a discrepancy.
  2. Page 2, 4th line from bottom: physical activity → PA
  3. Table 1, Page 5, 4th line: … compared with 54.1 (mL/kg/m²) in those… → … compared with 54.1 in those…
  4. Table 1, Pages 6, 7, and 8 each contain one typo: “Quemo”. “Chemo” is likely correct.
  5. Table 1, Page 8, van Beek et al. paper, outcomes of interest column, 8th row: oncovin (vincristine) → vincristine
  6. Table 1, Page 9, Wogksch et al. paper, outcomes of interest column, "↑ Muscular fatigue only in males vs. controls (adjusted mean difference of right knee work fatigue: 1.2 [99% CI: -8.9, 6.4] and left knee work fatigue: 4.2 [99% CI: -9.9, 1.6])." This appears to indicate no statistically significant difference. The same statement appears on page 10, section 3.3. Fitness, lines 7-9. Are these statements correct?
  7. Table 1, page 9, in the “outcomes of interest” column for Wogksch et al.'s paper, “↑ Body fat percentage only in males vs. controls (adjusted mean difference: 0.6 [99% CI: -2.7, 1.4])” appears to indicate no statistically significant difference. Is this statement correct? In the main text, on page 10, lines 3–4 of section 3.4. Body Composition, this paper is mentioned as follows: "Wogksch et al. [28] found similar body fat percentages among male (25.4 ± 0.6 vs. 24.8 ± 0.6%) and female (35.2 ± 0.7 vs. 35.1 ± 0.7%) survivors of HL and controls."
  8. Page 11, Section 3.6. Associations of treatment with poor fitness, physical inactivity, body composition and autonomic dysfunctions, line 20: oncovin (vincristine), → vincristine,

Author Response

Comments 1: This is an important paper addressing significant late complications resulting from radiation therapy and chemotherapy administered for Hodgkin lymphoma.

Response 1: Thank you.

Comments 2: The 13th line of the Abstract and the first line of Section 2.3. Screening data extraction on page 3 state that 2,527 papers were included, but Figure 1 shows 2,886 papers were extracted at first, creating a discrepancy.

Response 2: We apologise for this discrepancy. The search was updated at the end of September 2025 identifying a total of 2,886 manuscripts from all databases. This number has been updated.

Comments 3: Page 2, 4th line from bottom: physical activity → PA

Response 3: This edit has been made.

Comments 4: Table 1, Page 5, 4th line: … compared with 54.1 (mL/kg/m²) in those… → … compared with 54.1 in those…

Response 4: This edit has been made.

Comments 5: Table 1, Pages 6, 7, and 8 each contain one typo: “Quemo”. “Chemo” is likely correct.

Response 5: These edits have been made.

Comments 6: Table 1, Page 8, van Beek et al. paper, outcomes of interest column, 8th row: oncovin (vincristine) → vincristine

Response 6: This edit has been made.

Comments 7: Table 1, Page 9, Wogksch et al. paper, outcomes of interest column, "↑ Muscular fatigue only in males vs. controls (adjusted mean difference of right knee work fatigue: 1.2 [99% CI: -8.9, 6.4] and left knee work fatigue: 4.2 [99% CI: -9.9, 1.6])." This appears to indicate no statistically significant difference. The same statement appears on page 10, section 3.3. Fitness, lines 7-9. Are these statements correct?

Response 7: Thank you for the comment. The reviewer is correct. This has been amended accordingly. Muscular fatigue was not significantly higher only in males vs. controls. However, what authors mention is that closely to 20% of male survivors had excess impairment (compared to 6.7% expected in the general population) in muscular fatigue in comparison to age-, sex-, and race-specific comparison group values (percentage of those with > 1.5 SD). This proportion was less than 6.7% in female survivors. This test is included in Table 1 and in the text on page 10.

Comments 8: Table 1, page 9, in the “outcomes of interest” column for Wogksch et al.'s paper, “↑ Body fat percentage only in males vs. controls (adjusted mean difference: 0.6 [99% CI: -2.7, 1.4])” appears to indicate no statistically significant difference. Is this statement correct? In the main text, on page 10, lines 3–4 of section 3.4. Body Composition, this paper is mentioned as follows: "Wogksch et al. [28] found similar body fat percentages among male (25.4 ± 0.6 vs. 24.8 ± 0.6%) and female (35.2 ± 0.7 vs. 35.1 ± 0.7%) survivors of HL and controls."

Response 8: Thank you for the comment. As previously mentioned, the reviewer is right and therefore, this has been amended accordingly. Body fat percentage was not significantly higher only in males vs. controls. However, nearly 20% of male survivors had impaired body fat percentage in comparison to age-, sex-, and race-specific comparison group values (> 1.5 SD). This test is included in Table 1 and in the text on page 10.

Comments 9: Page 11, Section 3.6. Associations of treatment with poor fitness, physical inactivity, body composition and autonomic dysfunctions, line 20: oncovin (vincristine), → vincristine.

Response 9: This edit has been made.

Reviewer 3 Report

Comments and Suggestions for Authors

This is a scoping review address the association of physical activity, fitness, perceived health, and chronic conditions with mortality in adult survivors of childhood and young adult Hodgkin’s lymphoma. The topic is of interest, however, there are multiple issues in evidence synthesis and reporting.

  1. The Table 1 does not seem an appropriate fit for Method. Instead, the authors should clearly define the MESH term they use to search for "physical activity", "fitness", "perceived health", "chronic disease", and "mortality" (all-cause mortality or cancer specific mortality), and the definition of "adult survivors of childhood and young adult Hodgkin’s lymphoma" etc. The Methods need to be substantially improved in details.
  2. Table 1 is fit for Results section. However, the authors did not explain why they organized the study in the current sequence. It is not by publication year or any obvious categorization. I suggest the author should categorize the articles with "exposure variable" such as physical activity and use publication year to order the articles in each category. 
  3. The title states that this study focus on the influencing factors associated mortality in adult survivors of childhood and young adult Hodgkin’s lymphoma. However, I did not notice information about mortality in most citations they have extracted in the Table 1. The author should include another column to specify their reported association with the mortality.
  4. In the Discussion, the author did not have a paragraph explaining the reason why PA, perceived fitness, chronic health conditions are associated with mortality of childhood and young adult HL survivors, which is the main theme of this study. They author should put greater efforts in provide the explanation before jumping into the intervention part.

Author Response

Comments 1: This is a scoping review address the association of physical activity, fitness, perceived health, and chronic conditions with mortality in adult survivors of childhood and young adult Hodgkin’s lymphoma. The topic is of interest, however, there are multiple issues in evidence synthesis and reporting.

Response 1: Thank you. We have carefully considered all comments and addressed all issues in the manuscript.

Comments 2: The Table 1 does not seem an appropriate fit for Method. Instead, the authors should clearly define the MESH term they use to search for "physical activity", "fitness", "perceived health", "chronic disease", and "mortality" (all-cause mortality or cancer specific mortality), and the definition of "adult survivors of childhood and young adult Hodgkin’s lymphoma" etc. The Methods need to be substantially improved in details.

Response 2: Thank you for the suggestion. We have moved Table 1 to the results since it fits better in this section. The terms that we used in the search strategy were presented in the appendix A1 for each database. Our search strategy did not include MESH terms because they only exist in MEDLINE (via PubMed) database and hence, using them in other databases such as Web of Science, CINAHL and Cochrane is impossible and would not be a systematic search. Some studies could be retrieved using MESH terms in MEDLINE (via PubMed), but the same studies would not be identified in the rest of databases. Therefore, we decided to use the same terms across all databases. We have followed the Reviewer’s recommendation and hence, the terms that we used in the search strategy have also been included in the first paragraph of the methods.

Comments 3: Table 1 is fit for Results section. However, the authors did not explain why they organized the study in the current sequence. It is not by publication year or any obvious categorization. I suggest the author should categorize the articles with "exposure variable" such as physical activity and use publication year to order the articles in each category.

Response 3: Thank you for the comment. We have moved Table 1 to the results section, and it has been recategorized based on the exposure variable and publication year to order the articles in each category. Studies fitting in more than one category are only included in one of them to avoid misunderstandings with the total number of included manuscripts.

Comments 4: The title states that this study focus on the influencing factors associated mortality in adult survivors of childhood and young adult Hodgkin’s lymphoma. However, I did not notice information about mortality in most citations they have extracted in the Table 1. The author should include another column to specify their reported association with the mortality.

Response 4: Thank you very much for the recommendation. Unfortunately, only 10% (2/20) of the included studies evaluated mortality risk and hence, we have considered that adding another column would leave 18 studies with empty spaces. We now highlight in bold in table 1 the reported association with the mortality of the two mentioned studies (Jones et al.1 and Groarke et al.2).

Comments 5: In the Discussion, the author did not have a paragraph explaining the reason why PA, perceived fitness, chronic health conditions are associated with mortality of childhood and young adult HL survivors, which is the main theme of this study. They author should put greater efforts in provide the explanation before jumping into the intervention part.

Response 5: Thank you for the suggestion. We have accordingly modified the discussion so that a full paragraph now is focused on the associations with mortality, before discussing interventions.

REFERENCES

  1. Jones LW, Liu Q, Armstrong GT, Ness KK, Yasui Y, Devine K, Tonorezos E, Soares-Miranda L, Sklar CA, Douglas PS, Robison LL, Oeffinger KC. Exercise and risk of major cardiovascular events in adult survivors of childhood hodgkin lymphoma: a re-port from the childhood cancer survivor study. J Clin Oncol. 2014, 32, 3643-50. doi: 10.1200/JCO.2014.56.7511.
  2. Groarke JD, Tanguturi VK, Hainer J, Klein J, Moslehi JJ, Ng A, Forman DE, Di Carli MF, Nohria A. Abnormal exercise re-sponse in long-term survivors of hodgkin lymphoma treated with thoracic irradiation: evidence of cardiac autonomic dys-function and impact on outcomes. J Am Coll Cardiol. 2015, 65, 573-83. doi: 10.1016/j.jacc.2014.11.035.

Round 2

Reviewer 3 Report

Comments and Suggestions for Authors

Thanks for the revision to make the manuscript clear. I have no further comments.